# IL-15 and N-803 for HIV Cure Approaches

**DOI:** 10.3390/v15091912

**Published:** 2023-09-12

**Authors:** J. Natalie Howard, Alberto Bosque

**Affiliations:** Department of Microbiology, Immunology and Tropical Medicine, George Washington University, Washington, DC 20037, USA; jnhoward@email.gwu.edu

**Keywords:** HIV-1, latency, reservoir, shock and kill, IL-15, N-803

## Abstract

In spite of the advances in antiretroviral therapy to treat HIV infection, the presence of a latent reservoir of HIV-infected cells represents the largest barrier towards finding a cure. Among the different strategies being pursued to eliminate or reduce this latent reservoir, the γc-cytokine IL-15 or its superagonist N-803 are currently under clinical investigation, either alone or with other interventions. They have been shown to reactivate latent HIV and enhance immune effector function, both of which are potentially required for effective reduction of latent reservoirs. In here, we present a comprehensive literature review of the different in vitro, ex vivo, and in vivo studies conducted to date that are aimed at targeting HIV reservoirs using IL-15 and N-803.

## 1. Introduction

It has been more than 30 years since the development of antiretroviral therapy (ART) for the Human Immunodeficiency Virus (HIV), which has turned the disease from a death sentence into a manageable chronic condition. Current ART regimens consist of two to three drugs from different classes, which has significantly decreased the development of drug resistance compared to single drug administration [1]. The first comprehensive analysis of viral decay dynamics after treatment with ART determined that there is an initial steep decline (~99%) in plasma viremia, followed by a slower second phase of decay where the contributors are characterized as long-lived infected cells, including infected macrophages and T cells [2,3]. Mathematical modeling estimated a time of about 3 years for complete viral decay, and longer if virus was retained in additional sanctuary sites [2]. It was unknown whether these infected cells could persist in the presence of therapy, but this was subsequently answered by studies from three separate groups showing that replication-competent virus could be recovered from participants fully suppressed on ART [4,5,6]. These findings were the first evidence of the latent HIV reservoir, where infected cells do not actively produce viral proteins, allowing them to persist in the presence of therapy long-term. Latently infected cells were calculated to be present at extremely low frequencies in vivo [7], but were the source of rapid viral rebound upon interruption of ART [8]. Longitudinal analysis has shown that decay of long-lived latently infected memory CD4 T cells is very slow (t_1/2_~44 months) [3,9,10], and would require over 70 years of continuous ART to fully eradicate the reservoir [11]. Thus, while ART has altered the course of HIV infection, turning it from a death sentence into a manageable chronic disease, it is not curative and leaves a need for development of cure strategies that will be globally implementable.

One of the most highly investigated strategies to target and eliminate the latent HIV reservoir to date is the “shock and kill” strategy [12]. It involves using a latency-reversing agent (LRA), typically a small molecule compound, to induce viral gene expression from latently infected cells (“shock”). Once this occurs, infected cells are now visible to effector cells of the innate and adaptive immune system, such as natural killer (NK) cells and CD8 T cells, respectively, and can be killed by these or by the innate cytopathic effects of the virus (“kill”). The aim of this strategy is to eliminate all replication-competent virus. Multiple classes of LRAs have been developed to target and antagonize mechanisms known to promote viral latency. Among these, toll-like receptor (TLR) agonists, epigenetic modifiers, protein kinase C (PKC) agonists, non-canonical NF-κB agonists, and common γ chain cytokines (γc-cytokines) have been the most studied.

Regarding γc-cytokines, initial attempts at latency reversal following the identification of the latent reservoir involved Interleukin-2 (IL-2). However, severe side effects from toxicity were observed, including renal failure and seizures, and the clinical trials were halted [13,14,15,16]. The closely related IL-7 has also been shown to have LRA activity in vitro [17], but subsequent studies observed that IL-7 induces homeostatic proliferation and has minimal LRA activity [18,19]. In vivo studies have also shown that IL-7 may promote expansion of the latent reservoir [18,20,21,22]. This review will focus on the use of the γc-cytokine IL-15 for HIV cure interventions.

## 2. IL-15

IL-15 was identified in 1994 as a growth factor for T lymphocytes, primarily memory CD8 T cells and NK cells [23], and has been shown to be a vital factor in T cell division in vivo [24]. The IL-15 receptor is made up of three subunits: the unique α chain (IL-15Rα); the β chain, which is shared with the IL-2 receptor; and the common γ chain, which is shared with all the γc-cytokines (IL-2, -4, -7, -9, -15 and -21) [25]. IL-15 production can be induced through Type-I interferon (IFN) signaling in response to viral infections such as HIV [26,27]. It has been shown that activated monocytes and macrophages produce IL-15 [23], and this produced IL-15 binds to the high-affinity IL-15Rα typically on cells like dendritic cells (DCs) and stromal cells [28]. The IL-15/IL-15Rα complex can then either interact with the β/γc receptor complex on the same cell in a *cis*-manner, or interact with the β/γc receptor complex on target cells such as NK or T cells in a *trans*-manner [28,29]. Ligand-receptor binding in either manner results in signaling through activation of Janus Kinase 1 (JAK1) and JAK3, and further downstream activation of primarily signal transducer and activator of transcription 3 (STAT3) and STAT5 [30] (Figure 1).

IL-15 signaling, and subsequent STAT activation, are tightly regulated through several negative feedback mechanisms (Figure 1). Among these, suppressors of cytokine signaling (SOCS) mediate a negative feedback loop, which results in JAK dephosphorylation, leading to abrogation of STAT signaling. Of the eight members of the SOCS family, IL-15 has been shown to induce the cytokine-inducible SH2-containing protein (CIS) [31], SOCS2, and SOCS3 [32]. CIS directly interacts and dephosphorylates JAK1 [33], leading to abrogated STAT5 activation, SOCS2 directly binds to and dephosphorylates JAK2 [34], and SOCS3 has been shown to bind to and dephosphorylate both JAK1 and JAK2 [35]. The second main feedback loop is controlled by protein inhibitors of activated STAT (PIAS) proteins. The PIAS protein family is localized within the nucleus and directly associates with STAT proteins to constitutively repress activation [36,37]. Of the members of the PIAS family, only PIAS3 has been shown to associate with STAT5 [38]. The final, key negative feedback mechanism that regulates IL-15-induced STAT activation is mediated by protein tyrosine phosphatases (PTPs) [39,40]. The PTP superfamily is comprised of classical phosphotyrosine-specific phosphatases and dual-specific phosphatases (DSPs) [41,42]. Classical PTPs are further divided into receptor-like tyrosine phosphatases, which contain a transmembrane domain [43,44], and nonreceptor tyrosine phosphatases (NTPs), which do not [45]. STAT5 has been identified as a target of multiple PTPs, including the NTPs SHP2 [46,47,48], PTP1B (PTPN1) [49,50,51,52], and PTPN2 (TC-PTP) [53,54,55]. These enzymes dephosphorylate activated STAT proteins within the cytoplasm and nucleus, as opposed to upstream JAK proteins, leading to transcriptional repression.

In the context of HIV infection, endogenous IL-15 has been shown to promote expansion of CD8 T cells in ART-naïve people living with HIV (PWH), and lymph node IL-15 levels were found to correlate with circulating CD8 counts in untreated PWH compared to fully suppressed or HIV-negative individuals [56]. However, studies investigating the production of endogenous IL-15 during HIV infection have produced inconsistent results. It has been observed that serum IL-15 levels are reduced in ART-naïve PWH compared to HIV-negative controls [57], and that IL-15 production by PBMCs from ART-naïve PWH is significantly impaired compared to HIV-negative controls [57,58]. In contrast, several groups have observed significantly higher IL-15 levels in ART-naïve PWH [59,60,61], and these higher levels were shown to correlate with increased viral load [61,62]. On the other hand, other studies have shown higher levels of IL-15 in monocytes of long-term non-progressors [63], as well as higher IL-15 levels corresponding to better clinical responses to ART [58,64]. Overall, there appears to be significant variability in the effects of endogenous IL-15 during the course of natural HIV infection.

In addition to promoting survival and proliferation of NK cells and memory CD8 T cells [65,66,67], IL-15 has been shown to enhance activation of these cells, leading to increased production of the effector molecules IFN-γ and TNFα, enhanced killing capacity of target cells [68,69], and decreased expression of pro-apoptotic proteins [70]. For these reasons, IL-15 has been under investigation for treatment of cancer and viral infections, including HIV. However, soluble IL-15 has been shown to have low in vivo biological activity and a short half-life, prompting development of IL-15 agonist complexes. One superagonist in particular, N-803, has been tested in vitro and in vivo for anti-HIV effects.

## 3. N-803

Originally known as ALT-803, N-803 combines the highly potent IL-15 superagonist IL-15N72D [71] with the soluble domain of the IL-15Rα (IL-15RαSu) to form a fusion protein with 25 times the in vivo biological activity of soluble IL-15 [72,73] (Figure 1). N-803 has shown broad anti-tumor effects in several murine cancer models [74,75], as well as increased tissue retention compared to soluble IL-15 [76].

## 4. Anti-HIV Activity of IL-15/N-803 In Vitro and Ex Vivo

Several studies in vitro and ex vivo have evaluated the potential latency-reversal activity and ability to enhance immune effector functions of IL-15/N-803 in cell lines, primary cells, and cells from PWH.

Regarding latency reversal, initial work from Chehimi et al. observed minimal viral replication in the U1 and ACH2 chronically infected cell lines upon stimulation with IL-15, as well as minimal replication from αCD3/IL-15-stimulated PBMCs from PWH [77]. Conversely, Jones et al. showed increased viral p24 in supernatants from latently infected CD4 T cells in vitro and increased viral RNA in ex vivo CD4 T cell supernatants from ART-suppressed individuals after stimulation with both IL-15 and N-803 alone [78]. Additionally, reactivation of latent virus with IL-15 has also been observed in a different CD4 T cell model [79].

Regarding the immune modulatory effects of IL-15/N-803, IL-15 stimulation of PBMCs resulted in increased expansion [80], activation [81], and effector functions [80] of HIV-specific CD8 T cells. N-803 priming by targeted nanogel loading of an HIV-specific CD8 T cell clone was also observed to enhance proliferation and cytotoxic activity in vitro [82]. Additionally, ex vivo treatment of HIV-specific CD8 T cells from ART-suppressed PWH with N-803 enhanced the cytotoxicity as measured by IFNγ ELISPOT [83]. Similarly, IL-15 treatment of NK cells results in enhanced cytotoxicity [77,84,85,86], killing capacity of HIV-infected cells [79,84,85,87,88], production of IFN-γ [89], and antibody-dependent cellular cytotoxicity (ADCC) function [85,86,87,90].

## 5. Anti-HIV Activity of IL-15/N-803 in Animal Models

Non-human primate (NHP) studies have been conducted using both IL-15 and N-803 to investigate the impact of treatment on immune functions and latency reversal in a variety of different contexts, including in chronic infection in the absence of ART, in the presence of ART before and after analytical treatment interruption (ATI), and prior to ART administration.

### 5.1. IL-15 Studies

In work from Mueller et al., six chronically simian immunodeficiency virus (SIV)-infected cynomolgus macaques were treated with either low dose (10 μg/kg) or high dose (100 μg/kg) recombinant IL-15 twice weekly for four weeks. A 2.4-fold increase in total peripheral blood CD8 T cells was observed in the high-dose group compared to control animals with effector memory CD8 T cells being preferentially increased over other populations and sustained throughout the treatment period. However, this increase did not result in enhanced cytotoxicity. A non-significant increase in NK cell absolute counts was also reported in the high-dose group compared to controls, but cytotoxicity was not evaluated. Additionally, no decrease in viral load was seen for either treatment group, but no severe adverse side effects were reported even in the high-dose group [91]. In a follow-up study from the same group, six acutely SIV-infected rhesus macaques were treated with 100 μg/kg recombinant IL-15 twice weekly for 4 weeks. An increase in viral set point was observed in all IL-15-treated animals compared to those untreated, despite the presence of two- to three-fold more SIV-specific CD8 T cells and NK cells. Based on the observation that IL-15 increased CD4 T cell activation early after treatment, the authors hypothesized that IL-15-mediated CD4 T cell activation resulted in an increased viral set point that could not be controlled by either higher levels of CD8 or NK cells [92].

The effects of IL-15 administration in combination with ART in chronically SIV-infected rhesus macaques on immune reconstitution during therapy and after treatment interruption have also been studied [93]. In this study, three groups of animals were used: ART-alone, IL-15 alone (80 μg/kg twice weekly), and ART+IL-15. Animals treated with ART+IL-15 took, on average, 4 weeks to reach complete viral suppression compared to 3 weeks in the ART-alone group, and animals treated with IL-15 had no decrease in viral load. The presence of IL-15 had no effect on time to rebound when ART was interrupted after 46 days. Enhanced reconstitution of peripheral blood CD4 T cells was not observed in the ART+IL-15 group compared to the ART alone group; however, a transient increase in CD8 T cells was detected in the periphery in the IL-15-treated animals. IL-15 was found to mediate proliferation of antigen-specific CD8 T cells, but the quality of the response (IFN-γ production) was not increased compared to ART alone. Additionally, upon ART interruption, ART+IL-15-treated animals experienced a higher rate of CD4 T cell depletion compared to the ART-alone group [93]. However, work from the Pavlakis group investigating the effects of heterodimeric IL-15 (IL-15/IL-15Rα complex) in uninfected and SHIV-spontaneous-controlling rhesus macaques observed an increase in CD8 T cells expressing granzyme B in B cell follicles in the lymph nodes in both treated groups. In SHIV-infected animals, there was a general decrease in viral burden in both the lymph nodes and plasma [94].

### 5.2. N-803 Studies

Original proof-of-concept in vivo testing of N-803 in murine multiple myeloma models demonstrated that N-803 has enhanced serum half-life compared to soluble IL-15, and promoted rapid expansion of IFNγ-producing memory CD8 T cells [72,73]. Further murine safety studies measuring compound biodistribution found that N-803 was retained in lymphoid tissues better than soluble IL-15, had a more favorable pharmacodynamic profile, and was well tolerated during a weekly dosing schedule [76]. Recently, N-803 was shown to enhance the antiviral effects of an HIV-specific CD8 T cell clone when administered as a targeted nanogel in a participant-derived xenograft (PDX) model of HIV [82].

The effects of N-803 have been extensively studied in NHP models. Administration of N-803 in ART-naïve, chronically SIV-infected rhesus macaques was shown to transiently decrease viremia in the immediate days following treatment. This corresponded to an increase in CD8 T cell and NK cell absolute counts, but no enhanced cytotoxicity was observed from these effector cells. Plasma viremia rebound while still under N-803 treatment suggested that effector cells become refractory to cytokine stimulus, and it was shown that surface expression of IL-15 receptors CD122 and CD132 on central memory CD8 T and NK cells decreased over time during continuous N-803 treatment [95]. One important additional finding was the ability of N-803 to promote proliferation, activation, and migration of NK cells and virus-specific CD8 T cells to B cell follicles in chronically SIV-infected rhesus macaques. However, this did not correspond to a reduction in plasma viral load although fewer SIV RNA positive cells were observed in the lymph nodes following N-803 treatment [96]. In a follow up study from the Sacha group investigating the effects of N-803 on latency reversal in SHIV-infected, ART-suppressed rhesus macaques, it was once again observed that N-803 treatment resulted in enhanced proliferation and mobilization of NK cells and SHIV-specific CD8 T cells to lymph node B cell follicles. No latency-reversal activity was seen and there was no reduction in CD4 T cell-associated viral DNA levels before ART interruption between the N-803-treated animals and the controls [97]. In contrast, two studies from the Silvestri group have shown that N-803 can reactivate virus in vivo with concomitant CD8 T cell depletion. The first study examined the effects of using a CD8α-depleting antibody, which depletes NK cells, NKT cells, and gamma delta (γδ) T cells in addition to CD8 T cells, alone and in combination with N-803 administration in SIV-infected, ART-suppressed rhesus macaques. Reactivation was observed despite continued ART therapy in the animals treated with the combination of depletion + N-803; however, this had no effect on the size of the latent SIV reservoir measured before and after treatment interruption [98]. The second study utilized both a CD8α-depleting antibody and a CD8β-depleting antibody, which is specific for CD8 T cells alone, in combination with N-803 treatment in SHIV-infected, ART-suppressed rhesus macaques. The depletions were performed sequentially, with six months in between the CD8α and CD8β-depletions in order for sufficient CD8 reconstitution to occur. Levels of viral reactivation were observed in the combination-treated animals after CD8α depletion and N-803 administration, as was seen in the previous CD8α-depleting study [98], but reactivation levels following CD8β depletion and N-803 administration were found to be less robust. This was potentially attributed to the animals’ increased time on ART at the time of CD8β depletion compared to CD8α depletion (18 months vs. 12 months). Overall, CD8β depletion did not have a measurable effect on the size of the viral reservoir, and the levels of reactivation were found to be inversely correlated with the efficacy of CD8 T cell depletion [99].

## 6. N-803 Activity in Clinical Trials

The first clinical trial measuring the safety and efficacy of N-803 in humans was performed on patients experiencing hematologic malignancy relapse post allogeneic hematopoietic cell transplantation. It was observed that subcutaneous administration of N-803 resulted in prolonged serum concentrations compared to intravenous administration, along with increased expansion of NK and CD8 T cells. No treatment-related graft-versus-host disease symptoms were recorded, and N-803 was deemed safe and well tolerated in humans [100].

A phase 1 clinical trial investigating N-803 in PWH was published in early 2022. The main goal of the study was to confirm safety and tolerability of N-803 while secondarily assessing the impact on latent reservoir size. Dose escalation from 0.3–6.0 μg/kg was performed on four cohorts of ART-suppressed individuals, and three doses were given one week apart subcutaneously. N-803 was found to be well tolerated at all doses, with 6.0 μg/kg reaching the maximum dose threshold and no significant adverse effects being observed. Increased proliferation of CD4, CD8, and NK cells was observed regardless of dosage, and enhanced activation of CD8 and NK cells was also measured after N-803 administration. A sustained increase in absolute NK counts was observed from pre dose 1 to after the third dose, but this was not seen for CD8 T cells. N-803 administration did not influence degranulation of HIV-specific CD8 T cells ex vivo. Evaluation of the viral reservoir using the envelope detection by induced transcription-based sequencing (EDITS) assay [101] measured a transient increase in viral transcripts from pre dose 1 to 7 days post dose 3, which had decreased to just above pre-dose 1 baseline levels at the 6-month post-dose 3 timepoint in unstimulated PBMCs. In ex vivo stimulated PBMCs, there was a steady decrease in viral transcripts that was sustained over 6 months post final N-803 administration, regardless of dosage. Intact proviral DNA (IPDA) analysis showed an increase in proviral DNA copies between the pre-dose 1 and post-dose 3 timepoints, but there was no change in the number of cells harboring 5′ or 3′ defective provirus [102]. Overall, this study demonstrated that N-803 is safe and tolerated in PWH, but no clear impact of N-803 was observed on the latent viral reservoir, similar to the NHP studies discussed above.

## 7. Future Directions of IL-15/N-803 in HIV Cure Studies

Overall, there is promise in utilizing IL-15 and/or N-803 as a component of HIV cure strategies. It has been shown that N-803 enhances proliferation and activation of NK cells and CD8 T cells in ART-suppressed PWH in vivo [102], enhances migration of virus-specific CD8 T cells and NK cells from peripheral blood to B cell follicles [96,97], and might be an important component in facilitating clearance of tissue reservoirs during complete ART viral suppression [103].

As a latency-reversing agent, N-803 has been shown to have a minor and transient effect on HIV reactivation in vivo when used as a monotherapy [102]. Currently, there are two clinical trials in the recruiting phase that will test the ability of N-803, with and without broadly neutralizing antibodies (bNAbs), to control infection during analytical treatment interruption (NCT04340596 and NCT05245292). These studies will also measure the impact on HIV-specific CD8 T cell and NK cell functional responses during the intervention period, as well as the effect on reservoir composition and size and the time to viral rebound. Given the ex vivo results of N-803 enhancing ADCC [85,86,87], it can be hypothesized that the combination of N-803 and bNAbs will lead to increased clearance of reactivated or productively infected cells through antibody-dependent mechanisms.

However, all these previous studies have highlighted the insufficient activity of IL-15 or N-803 alone. Therefore, further studies are needed to address the impact of combination strategies with IL-15 and additional therapeutics on the latent viral reservoir for optimal formulation of future treatment approaches. We have previously identified the small molecule 3-Hydroxy-1,2,3-benzotriazin-4(3H)-one, HODHBt, which is able to reactivate latent virus by enhancing STAT5 activation and binding to the HIV LTR [104]. Additionally, HODHBt, in combination with IL-15, enhances reactivation of latent virus ex vivo from PWH and enhances immune effector functions of NK cells and CD8 T cells against HIV-infected cells [88,105,106]. We also showed that single administration ex vivo of a combination of IL-15 and HODHBt can lead to a reduction in intact proviruses measured using IPDA in a subset of participants [106]. Recently, we have demonstrated that HODHBt binds and inhibits the catalytic domain of the non-receptor tyrosine phosphatases PTPN1 (also known as PTP1B) and PTPN2 [107]. Several new small molecules with subnanomolar activity that target PTPN1 and PTPN2 have been developed to target this pathway. ABBV-CLS-484 and ABBV-CLS-579 are two potent PTPN1/PTPN2 inhibitors, developed by Calico Life Sciences LLC, currently in phase I clinical trials being tested for solid tumors, either alone or in combination with other therapeutics (NCT04417465, NCT04777994). In a recent study, a related compound of the PTPN1/PTPN2 inhibitor ABBV-CLS-484, Compound-182, showed promise in small animal models for cancer therapeutics [108]. This study shows that administration in vivo of Compound-182 was associated with enhanced activation and recruitment of T cells in solid tumors, leading to a reduction in tumor growth. Most importantly, this was achieved in the absence of the development of cytokine release syndrome or autoimmunity, suggesting that targeting PTPN1 and PTPN2 in vivo may not be associated with overt immune-related toxicities. Evaluation of these compounds in HIV cure approaches when using IL-15/N-803 could lead to novel combination strategies to promote meaningful reductions of the latent reservoir resulting in ART-free remission or a cure.

## Figures and Tables

**Figure 1 viruses-15-01912-f001:**
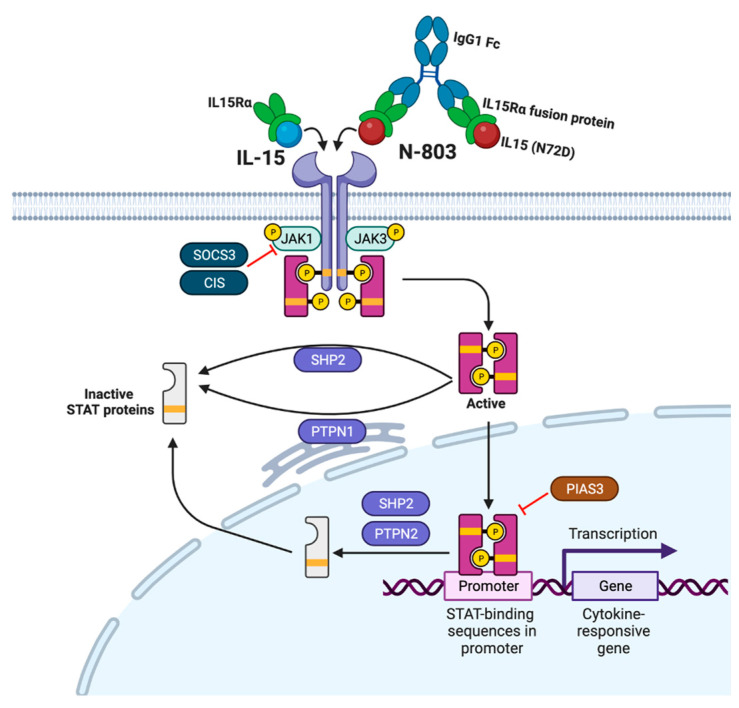
Signaling of IL-15 and N-803. IL-15 and N-803 signal through the same JAK/STAT pathway, resulting in activated STAT dimers binding to promoters and inducing gene transcription. Adapted from “Cytokine Signaling through the JAK/STAT pathway” (2023). Created with BioRender.com.

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
