# Peer review of "IL-15 and N-803 for HIV Cure Approaches"

_viruses, 2023, doi:10.3390/v15091912_

Round 1

Reviewer 1 Report

Howard and Bosque have written a very timely and comprehensive review of the role of IL-15 and N-803, an IL-15 superagonist in HIV cure strategies. Overall, this is an excellent manuscript, with very few faults.

One minor complaint – on lines 124-125 the authors say “HIV-positive individuals”. As they define PWH to mean “people living with HIV” just above on line 121, I’d recommend the authors use the abbreviation PWH instead of “HIV-positive individuals”.  

A second minor complaint – the headings for both IL-15 (line 58) and N-803 (line 111) are given as #2. Either these should be 2A and 2B, or N-803 should be #3, and the following headings renumbered.

One somewhat larger concern. There is a considerable body of literature about the role of IL-15 in the natural history of HIV infection, particularly the role of IL-15 in CD8 T cell activation and expansion. Little of that literature seems to be addressed here, and the authors should consider adding a small paragraph (or just a couple sentences) to the “2. IL-15” section describing some of the known roles of endogenous IL-15 in vivo in PWH.

Author Response

Howard and Bosque have written a very timely and comprehensive review of the role of IL-15 and N-803, an IL-15 superagonist in HIV cure strategies. Overall, this is an excellent manuscript, with very few faults.

“We will like to thank the reviewer for taking time to review our manuscript and provide constructive feedback. We hope that we have now addressed all the reviewers concerns”

One minor complaint – on lines 124-125 the authors say “HIV-positive individuals”. As they define PWH to mean “people living with HIV” just above on line 121, I’d recommend the authors use the abbreviation PWH instead of “HIV-positive individuals”.  

“We apologize for the use of improper language and we have now changed the sentence”

A second minor complaint – the headings for both IL-15 (line 58) and N-803 (line 111) are given as #2. Either these should be 2A and 2B, or N-803 should be #3, and the following headings renumbered.

“We have now change the headings”  

One somewhat larger concern. There is a considerable body of literature about the role of IL-15 in the natural history of HIV infection, particularly the role of IL-15 in CD8 T cell activation and expansion. Little of that literature seems to be addressed here, and the authors should consider adding a small paragraph (or just a couple sentences) to the “2. IL-15” section describing some of the known roles of endogenous IL-15 in vivo in PWH.

“We apologize for our lack of inclusion of the role of endogenous IL-15 in PWH in our previous manuscript. We have now included a paragraph in section 2. We hope that we have addressed the reviewers concerns”

Reviewer 2 Report

The authors present a comprehensive and well-written paper on the potential of IL-15 and N-803 in killing latent viral reservoirs. A concise, direct, and comprehensive discussion of the topic is presented. The addition/removal of additional chapters/topics is not required. However, adding these seven references is required for more complete information; this information should be added in those chapters or paragraphs the author considers most pertinent.

This cytokine is constitutively expressed in various cells, including macrophages, DCs, hepatocytes, epithelial, endothelial, and tumor cells (DOI: 10.1038/83253).

IL-15, alone or in combination with IL-12, significantly increased NK cell activity and antibody-dependent cellular cytotoxicity on mononuclear cells derived from HIV+ individuals (DOI: 10.1006/clin.1996.4298).

This cytokine induces IFN-γ and RANTES production in NK cells isolated from HIV viremic and aviremic patients (DOI: 10.1016/j.imlet.2005.10.001).

IL-15 levels in the serum of immunocompromised patients are significantly lower than in healthy subjects (DOI: 10.1023/a:1022568626500).

In human monocytes, the expression of this cytokine was significantly higher in patients with relative resistance to immune impairment caused by HIV (DOI: 10.1089/AID.2010.0317).

Increased production of IL-15 has been linked to better clinical responses to therapy, including co-infected patients (DOI: 10.1097/00002030-200201250-00006 and DOI: 10.1016/j.heliyon.2023.e15055).

Author Response

The authors present a comprehensive and well-written paper on the potential of IL-15 and N-803 in killing latent viral reservoirs. A concise, direct, and comprehensive discussion of the topic is presented. The addition/removal of additional chapters/topics is not required. However, adding these seven references is required for more complete information; this information should be added in those chapters or paragraphs the author considers most pertinent.

This cytokine is constitutively expressed in various cells, including macrophages, DCs, hepatocytes, epithelial, endothelial, and tumor cells (DOI: 10.1038/83253).

IL-15, alone or in combination with IL-12, significantly increased NK cell activity and antibody-dependent cellular cytotoxicity on mononuclear cells derived from HIV+ individuals (DOI: 10.1006/clin.1996.4298).

This cytokine induces IFN-γ and RANTES production in NK cells isolated from HIV viremic and aviremic patients (DOI: 10.1016/j.imlet.2005.10.001).

IL-15 levels in the serum of immunocompromised patients are significantly lower than in healthy subjects (DOI: 10.1023/a:1022568626500).

In human monocytes, the expression of this cytokine was significantly higher in patients with relative resistance to immune impairment caused by HIV (DOI: 10.1089/AID.2010.0317).

Increased production of IL-15 has been linked to better clinical responses to therapy, including co-infected patients (DOI: 10.1097/00002030-200201250-00006 and DOI: 10.1016/j.heliyon.2023.e15055).

“We will like to thank the reviewer for taking time to review our manuscript and provide constructive feedback. We have added the mentioned references and we hope that we have now addressed all the reviewer’s concerns”